# The Lipid Profile of the *Endomyces magnusii* Yeast upon the Assimilation of the Substrates of Different Types and upon Calorie Restriction

**DOI:** 10.3390/jof8111233

**Published:** 2022-11-21

**Authors:** Yulia I. Deryabina, Anastasia S. Kokoreva, Olga I. Klein, Natalya N. Gessler, Elena P. Isakova

**Affiliations:** Bach Institute of Biochemistry, Research Center of Biotechnology of the Russian Academy of Sciences, 119071 Moscow, Russia

**Keywords:** *Endomyces magnusii*, yeast, calorie restriction, metabolic readjustments, lipids

## Abstract

The study analyzes the dynamics in the lipid profile of the *Endomyces magnusii* yeast during the long-lasting cultivation using the substrates of “enzymatic” or “oxidative” type. Moreover, we studied its changes upon calorie restriction (CR) (0.5% glucose) and glucose depletion (0.2% glucose). Di-(DAGs), triacylglycerides (TAGs) and free fatty acids (FFAs) dominate in the storage lipid fractions. The TAG level was high in all the cultures tested and reached 80% of the total lipid amount. While being cultured on 2% substrates, the level of storage lipids decreased at the four-week stage, whereas upon CR their initially low amount doubled. Phosphatidylethanolamines (PE), sterols (St) (up to 62% of total lipids), phosphatidylcholines (PC), and phosphatidic acids (PA) (more than 40% of total lipids) were dominating in the membrane lipids of *E magnusii*. Upon CR at the late stationary growth stages (3–4 weeks), the total level of membrane lipid was two-fold higher than those on glycerol and 2% glucose. The palmitic acid C16:0 (from 10 to 23%), the palmitoleic acid C16:1 (from 4.3 to 15.9%), the oleic acid C18:1 (from 23.4 to 59.2%), and the linoleic acid C18:2 (from 10.8 to 49.2%) were the dominant fatty acids (FAs) of phospholipids. Upon glucose depletion (0.2% glucose), the total amount of storage and membrane lipids in the cells was comparable to that in the cells both on 2% and 0.5% glucose. High levels of PC and sphingolipids (SL) at the late stationary growth stages and an increased PA level throughout the whole experiment were typical for the membrane lipids composition upon the substrate depletion. There was shown a crucial role of St, PA, and a high share of the unsaturated FAs in the membrane phospholipids upon the adaptation of the *E. magnusii* yeast to the long-lasting cultivation upon the substrate restriction is shown. The autophagic processes in some fractions of the cell population provide the support of high level of lipid components at the late stages of cultivation upon substrate depletion under the CR conditions. CR is supposed to play the key role in regulating the lipid synthesis and risen resistance to oxidative stress, as well as its possible biotechnological application.

## 1. Introduction

Yeast cell lipids are valuable compounds performing numerous functions in the processes of compartmentalization, energy storage, and enzymes induction. Moreover, the compounds are an integral part of the plasma membrane responsible for its structure support, facilitating the solutes transport inside and outside the cell and cellular signaling necessary for cell survival. The *Saccharomyces* yeast, due to its highly conservative mechanisms common to all eukaryotes and some other advantages, has been well studied and used as a model for lipid research, which allowed some novel discoveries in this area. The construction of the mutant yeast deficient in some lipid metabolism reactions proved a most useful tool to studying biosynthesis of lipids, their degradation, and storage [1]. In the *Saccharomyces* yeast, like in the other eukaryotes, the regulation of lipid metabolism is extremely complicated. It involves the relationship between biosynthesis and mutual transformations of lots of lipid classes [2,3]. All eukaryotic cells have a coordination system of complex and interconnected lipid exchange pathways in different compartments of the cell upon the changes in availability of the precursors and membrane expansion during the cell division and growth. Genes, enzymes, and lipid metabolism pathways in the yeast are of significant homology with those in the higher eukaryotes that makes yeast a prospective model for biomedical and biotechnological study [4].

The *Endomyces magnusii* yeast are mesophilic, obligately aerobic yeast-like fungi capable of fermenting glucose, sucrose, raffinose, and galactose. Upon culturing in liquid media under the aerobic conditions, the yeast can assimilate glucose, sucrose, raffinose, galactose, *L*-sorbose, glycerol, *D*-mannitol, *D*-sorbitol, and succinate [5]. The *Endomyces magnusii* species was first isolated from oak slime and described by F. Ludwig in 1886. Later, the yeast were identified as *Magnusiomyces ludwigii*, *M. magnusii*, *Dipodascus magnusii*, and now, the names are considered synonymous. According to the modern classification, the *E. magnusii* yeast belongs to the *Saccharomycetes* (Saccharomycetales), the *Dipodascaceae* family (Dipodascaceae), the genus of *Magnusiomycetes* (Magnusiomyces) [6]. The *E. magnusii* yeast are multinucleated cells (Figure 1A,D) capable of forming pseudomycelia and true mycelium (Figure 1B) having septa with multiple micropores, and the main branches of their hyphae are 7–12 microns wide [7,8]. This type of the yeast is able to utilize efficiently both glucose, being a fermentable substrate, and glycerol, being an oxidative substrate. The study of the yeast mitochondria respiration revealed the full set of respiratory chain proteins, the three couple points upon oxidizing NAD-dependent substrates. It also showed no glucose repression accompanied by high metabolic mitochondria activity (Figure 1C) [9].

The composition of soluble cytosol carbohydrates in the yeast may include glucose and fructose, as well as polyols, namely arabitol, mannitol, trehalose, inositol, glycerol [10], accumulated in the cell vacuoles (Figure 1D,E). All this makes *E. magnusii* an attractive object for biotechnology, however, the data on this yeast application in biotechnological processes are quite limited. Thus, the study of the fumarate bioconversion [11,12] reported that fumarase activity in the disintegrated *E. magnusii* cells is about ten-fold higher than that in the disintegrated *Saccharomyces cerevisiae* cells, which can be a prerequisite for the *E. magnusii* application as a successful producer of *L*-malic acid. Upon growing on a synthetic YNB medium without yeast extract, the wild strain of *E. magnusii* is reported to produce abundant isobutanol (about 0.4 g/L) that is 10–20 times more than that in *S. cerevisiae*. Moreover, there was designed a technique, which permits to express the heterologous *IGF2* gene of *S. cerevisiae* encoding the acetolactate synthase into the *E. magnusii* yeast. This approach to the accumulation of more than 0.7 g/L of isobutanol. All the data clearly show that *E. magnusii* is of great potential as a butanol and *n*-butanol producer [13].

In the streamlined synthesis of biotechnologically significant compounds, the type of substrate applied in technological schemes, and, consequently, the metabolism type determined by it is of great importance. Thus, glucose is a favorable substrate for most yeast with metabolism of the “fermentative” type, in particular, the popular yeast object of *S. cerevisiae*. Thus, upon utilizing the carbohydrate, the expression of the genes responsible for the metabolism of other carbon sources is suppressed [14]. Furthermore, the yeast is capable of effective aerobic growth on glycerol, and the transition from “fermentation” to “respiratory” growth type requires a significant transcriptional rearrangement [15]. It includes the genes encoding the respiratory chain, oxidative, osmotic, and general anti-stress responses, protein synthesis enzymes, glycerol utilization, etc. [14]. The mechanisms of yeast adaptation to various types of metabolism have been extensively studied. 176 mitochondrial proteins of the *S. cerevisiae* yeast are reported to undergo the changes upon transiting from the fermentation metabolism type (glucose, galactose) to the respiratory one (lactate) [16]. Upon growth in the conditions of changing metabolism type (glucose fermentation, tricarboxylic acid cycle (TCA cycle), pyruvate dehydrogenase reaction) and the promoted respiratory chain, an increase in the phosphorylation degree was noted for eight proteins (Ald4p, Eft1p/2p, Eno1p, Eno2p, Om14p, Pda1p, Qcr2p, Sdh1p) [17]. The transcriptional regulation of various types of metabolism in yeast has also been studied in detail [18]. Roberts and Hudson [14] showed that the product of the *RSF1* gene, Rsf1p, can serve as a transcriptional modulator regulating the metabolic transition from glucose assimilation to glycerol one growth in *S. cerevisiae*. Rsf1p seems to coordinate the expression level of most genes involved in biological processes of both nuclear and mitochondrial gene compartments [14].

The regulation of the calories amount consumed also plays an essential role in changing the intracellular synthesis of compounds. The calorie restriction (CR) is a diet where only calorie intake is limited, while the intake of other nutrients (amino acids, vitamins, trace elements, etc.) is not limited [19]. This is the most studied and reproducible non-genetic intervention, which, as it has been confirmed by numerous studies, increases lifespan, and improves the organism health at various levels of organization: from unicellular yeast to primates and humans, indicating the involvement of conservative mechanisms universal for all eukaryotes in this process [20,21]. In our studies, the metabolic profile of *E. magnusii* was studied upon long-lasting cultivation on glycerol and glucose as sole carbon sources [22]. Our results revealed a high redox potential in the late and deep stationary growth phases in glycerol-utilizing cells. Moreover, we observed a high survival rate of the yeast cells using glycerol, while in the glucose-utilizing cultures, it decreased significantly in 7 days of growth. The study of the soluble cytosol carbohydrates profile of the E. *magnusii* yeast upon CR showed that cultures using 2% glucose and 1% glycerol contained mainly arabitol and mannitol. However, at low glucose concentrations they were substituted by inositol [10].

In this study, we consider the role of CR and calorie depletion in regulating the synthesis and lipid composition upon long-lasting cultivation of the *E. magnusii* yeast using the substrates of fermentable and oxidative types and a possible biotechnological application of this phenomenon.

## 2. Materials and Methods

### 2.1. Yeast Strains and Growth Conditions

*E. magnusii* yeast strain VKM U-261 obtained from the collection of the Institute of Biochemistry and Physiology of Microorganisms of the RAS. The *E. magnusii* yeast was grown in batches of 100 mL in glycerol—(1%) and glucose—(0.2%; 0.5%; 2%)—containing media of the following composition (g/L): MgSO_4_—0.5, (NH_4_)_2_SO_4_—0.3, KH_2_PO_4_—8.6, NaCl—0.1, CaCl_2_—0.05, yeast extract—2.0, *L*-histidine—2.75 mg, *L*-methionine—2.75 mg, and *L*-tryptophan—2.75 mg at 28 °C as described previously [20]. Absorbance was assessed in cell suspension at the wavelength of 590 nm (A_590_) using a Specol-11 spectrophotometer (Carl Zeiss, Oberkochen, Germany). Cells were harvested at different stages of growth: logarithmic (18.5 h of growth, A_590_ = 2.6–2.7), late stationary (48 h of growth, A_590_ = 4.5–4.6), deep stationary 1 (1 week of growth, A_590_ = 4.4–4.7), deep stationary 2 (2 weeks of growth, A_590_ = 4.4–4.7), deep stationary 3 (3 weeks of growth, A_590_ = 4.4–4.7), and deep stationary 4 (4 weeks of growth, A_590_ = 4.4–4.7).

### 2.2. DAPI Staining

Cellular nuclei were visualized using staining of cellular chromatin by DAPI with a subsequent analysis using an Axioskop 40 FL fluorescence microscope (Carl Zeiss, Oberkochen, Germany). The cells were raised in different growth phases, fixed, and permeabilized in 70% (*v*/*v*) ethanol. Yeast cells were centrifuged at 4000× *g* for 10 min, resuspended in phosphate saline buffer (PBS) containing DAPI (50 μg per mL), mixed gently, and incubated in the darkness at room temperature for 15 min. The cells visualized with a fluorescence microscope (at 100×), using a blue UV filter.

### 2.3. Potential-Dependent Staining

Potential-dependent staining of mitochondria in the *E. magnusii* cells raised in the logarithmic growth phase by rhodamine 123 (Rh123). Cells were incubated with 0.5 μM Rh123 and examined at 0, 15, 20, and 30 min. Incubation medium contained 0.01 M PBS, pH 7.4; 1% glycerol or glucose, respectively. Regions of high mitochondrial polarization are indicated by red fluorescence due to the concentrated dye. To examine the Rh123-stained preparations, filters 02 and 15 (Zeiss, Oberkochen, Germany) were used (magnification ×100). Photos were taken using an AxioCam MRC camera.

### 2.4. Transmission Electron Microscopy (TEM)

TEM analysis of untreated *E. magnusii* yeast cells was performed as described previously [23]. Briefly, the yeast cells were raised in the logarithmic or stationary (24 h) growth phase, precipitated, fixed with 2.5% glutaraldehyde in 0.1 M sodium phosphate buffer (pH 7.2) for 2 h, and then post-fixed in 1% OsO_4_ for an hour at room temperature. After dehydration, the samples were embedded in Epon 812. Ultrathin sections were prepared with an LKB-8800 ultratome using diamond knives. Thereafter, the sections were stained with uranyl acetate for 60 min, post-stained as described previously, and examined with a Jeol (JEM-100B) and Hitachi U-12 electron microscopes (Hitachi, Tokyo, Japan).

### 2.5. Preparation and Analysis of Lipids

To determine cell lipids, yeast cells in the stationary growth phase were raised and centrifuged at 6000× *g*, washed twice with cold distilled water, and frozen in liquid nitrogen [24]. The weighted sample of about 1 g was immediately homogenized in isopropanol to de-activate lipases by pestle and mortar, and incubated at 70 °C for 30 min. Then, the biomass was homogenized once more using some sand and the lipids were extracted by the method described in [24], which involved extraction with isopropanol and the isopropanol–chloroform mixture (1:1 and 1:2) at 70 °C, evaporation in a rotary evaporator, and extraction of the residue with chloroform-methanol (1:1) supplemented with 5% sodium chloride solution and water to remove water-soluble substances. After separating the mixture with a vortex, the chloroform layer was dried by passing it through water-free sodium sulfate, evaporated, and desiccated with a vacuum pump. The resulting pellet dissolved in a small amount of chloroform-methanol (1:1) was stored at −21 °C. The composition of storage lipids was assayed using an ascending thin layer chromatography on glass plates with silica gel 60 (Merck, Darmstadt, Germany). To separate storage lipids, the hexane/sulfuric ether/acetic acid (85:15:1) system [25] was used. To separate phospholipids (PL) and sphingolipids (SL), SI60 Silica thin layer chromatography plates were activated and developed in two dimensions, first with chloroform/methanol/water (65:25:4, by volume) and second with chloroform/acetone/methanol/acetic acid/water 950:20:10:10:5, by volume) [26]. The lipids (100–200 μg) were applied to a plate bovine serum glycoceramides, and phosphatidylcholine (PC) used as SL and PL standards were applied on chromatograms before their passing in the second direction. Samples of SL (5 and 10 µg) and PC (10 and 20 µg) were applied on the plates. To develop the stains, the chromatograms were sprayed with 5% sulfuric acid in ethanol, followed by heating up to 180 °C. To detect PL, the developed thin-layer plates were sprayed with 0.1% (*w*/*v*) ninhydrin for the ones carrying free amino groups and with *α*-naphtol for glycolipids. Lipids containing a quaternary ammonium group were visualized at room temperature with Dragendorff’s spray reagent [25]. SL were detected in the glycolipid fraction by the saponification method [25]. Storage lipids were identified with individual markers for di- and triglycerides (DAGs, TAGs), sterols (ergosterol) (St), free fatty acids (FFAs), and hydrocarbons (Sigma, St. Louis, MO, USA). Quantitative analysis of the lipids was performed using the Dens software (Lenchrom, Saint Petersburg, Russia). To assess the fatty acid composition of PL, separate PL were isolated using chromatography with two plates, eluted with chloroform/methanol (1/1, *v*/*v*) for a night. Then, the supernatant was decanted, evaporated, 1mL toluene and 2 mL of 2.5% H_2_SO_4_ dissolved in methanol and kept for two hours at +70 °C. Fatty acid methyl ethers were extracted with hexane, dried, and analyzed by a Kristall 5000.1 gas chromatograph (Chromatek, Yoshkar-Ola, Russia) using an Optima-240 (60 m × 0.25 mm) capillary column (Macherey-Nagel GmbH & Co., Duren, Germany). The temperature program was set from +130 °C to +240 °C. Eluting fatty acids were identified using the Supelco 37 Component FAME Mix (a mixture of fatty acid methyl esters) (Supelco, St. Louis, MO, USA).

### 2.6. Detection of the Reactive Oxygen Species ROS

The dynamics of intracellular ROS production was monitored using the spectroscopic fluorescence probe of dihydro-2′,7′-dichlorofluorescein diacetate ester (H_2_DCF-DA) (Sigma, Sant Louis, MO, USA). Upon crossing the cell membrane H_2_DCF-DA is hydrolyzed to a fluorescence species that remains trapped intracellularly, like its oxidation product. A stock solution of 8 mM H_2_DCF-DA in dimethyl sulfoxide (DMSO) was added to a cellular suspension to final concentrations of 40 μM H_2_DCFDA and 0.1% DMSO. The suspension was incubated in PBS in the dark for 30 min in medium containing 1.2 M sorbitol, 10 mM HEPES-buffer, 0.5% BSA, pH 7.4. The cells were then rinsed in the same medium and after 5–7 min the intensity of 2′,7′-dichlorofluorescein (DCF)-induced fluorescence was detected every 10 min for 2 h using dual wavelength photometry at 485 (λ_ex_)/528 (λ_em_) nm with a Synergy 2 (Bio Teck, Winooski, VT, USA) microplate photofluoriometer at a relative sensitivity of 100.

### 2.7. Statistical Analyses

Data are presented as the average ± the Standard Deviation in biological triplicates with a standard error of less than 5%. Analysis of soluble carbohydrates and lipids was performed using one-way ANOVA (n = 3). *p* values were determined by the two-tailed paired t-test at the 5% level of probability.

## 3. Results

### 3.1. Storage Lipids Profile

#### 3.1.1. Comparative Analysis of the Storage Lipids upon Culturing the *E. magnusii* Cells on 2% Glucose and 2% Glycerol (2% Glucose vs. 2% Glycerol)

Figure 2 shows the changes in the total composition and neutral (storage) lipids in the *E. magnusii* cells. The highest total amount of the storage lipids was found in the culture of 2% glycerol (Figure 2C): it reached its maximum at the three-week stage, then, in another week, it sharply 5-fold declined (Figure 2C). During glucose metabolism, the amount of the storage lipids varied significantly depending on the growth stage and carbohydrate concentration (Figure 2B). Thus, when cultivating yeast on 2% glucose containing medium, there was rather high initial content of storage lipids (34.07 mg/g *w*/*w*), followed by a twice decrease at the fourth week stage of cultivation. Analysis of the qualitative composition of the storage lipids of the culture demonstrated that DAGs, TAGs, and FFAs dominated in the cells (Figure 3). In the cultures grown on 2% glucose and 2% glycerol the TAGs level was high in all the cell samples and reached 80% of the total lipid content (Figure 3). Its amount transitively dropped to 13% in glycerol-utilizing cells at the one-week growth stage, followed by an increase at the two-week stage of cultivation (Figure 3B,C). At the same time, the cells using 2% glucose TAGs (50–70%) kept their high level up to 2 weeks of cultivation (Figure 3A–C), then decreased by the three-week growth stage and stabilized by the end of the experiment at a level of 44% (Figure 3D,E). The DAGs amount reflected no significant difference between the cultures grown on 2% substrates (Figure 3). The level of FFAs in the cells using glycerol increased by more than 5 times during the first week of cultivation (Figure 3B), followed by a gradual decrease to the initial level (24 h of growth) by the four-week stage (Figure 3E). By contrary, when using 2% glucose, it reached the maximum by 3–4 weeks of cultivation, which made up 30–35% (Figure 3D,E).

Sterol esters (Est) were minorities in the storage lipids, their share in the cellular lipids exceeded no more than 10–15% (Figure 3). The culture grown on 2% glucose at the three-week stage where Est reached 30% was the only exception, whereas that grown on 2% glycerol had the trace amounts of Est (1.5%) (Figure 3D). It should be emphasized that beginning from the one-week growth stage, the *E. magnusii* cells showed some unidentified components of lipid nature (Y1 and Y2 in Figure 3), in some cases their share made up 15%.

#### 3.1.2. The Storage Lipid Profile of the *E. magnusii* upon CR

In the cells using 0.5% glucose the level of storage lipids varied at the same level until the three-week stage then it doubled by the four-week stage compared to that at the beginning of the experiment (24 h of cultivation) (Figure 2A). Upon CR at the one-week stage, the TAGs amount steadily decreased accompanied by an increase in FFAs levels, which at the one-week stage significantly decreased, followed by a further increase to the initial level (Figure 3). The DAGs and Est levels fluctuated between 15–25% and 7–15%, respectively, throughout the whole experiment (Figure 3). It is of interest that the DAGs amount (about 25%) found in this variant at the 24 h growth stage and the FFAs level at the four-week stage were the highest of all the samples tested (Figure 3A,E). In the cells grown under the CR conditions, there was an unidentified lipid compound Y1 and its share in the total storage lipids reached nearly 9% at the one-, two-, and four-week stage (Figure 3B,C,E), and it increased up to 15% at the three-week growth stage (Figure 3D).

### 3.2. Membrane Lipids Profile

#### 3.2.1. Comparative Analysis of the Membrane Lipid Profile upon Cultivating the *E. magnusii* Yeast on 2% Glucose and 2% Glycerol (2% Glucose vs. 2% Glycerol)

The assay of the quantitative composition of the membrane lipids showed that the cells using 2% glucose at the one-week stage had the maximum amount of the components (41.750 mg/g, *w*/*w*) (Figure 4B). The culture using 2% glycerol displayed the peak in the membrane lipids amount at the one-two week stages, followed by its decrease by the end of the experiment (Figure 4A). As for the culture grown on 2% glucose, it showed the similar trend, except for the decrease in the membrane lipids amount even by the second week of the cultivation (Figure 4C).

PE, St (more than 30% of the total amount of lipids), PC and phosphatidic acids (PA) (more than 40%) were the most important components of the membrane lipids of the *E. magnusii* cultures (Figure 5). Among the minorities reaching no more than 10% of the total content, cardiolipins (CL), lysophosphatidylethanolamine + phosphatidylinositols (LPA + PI), and SL made the greatest contribution (Figure 5). The PE level remained at consistently high level for two weeks of cultivation in all the versions, followed by a more than two-fold decrease at the three- and four-week growth stage in the cells using 2% glucose (Figure 5D,E). The PC level also decreased in the culture using the substrate of the fermentative type for 1-2 growth weeks, followed by its stabilization by the end of the experiment (Figure 5). The PA amount reached its peak at the one-two-week stage of cultivation, collapsing in three weeks and almost disappeared by the four-week growth stage in both cultures using 2% glucose and 2% glycerol (Figure 5). The St amount nearly doubled at the three-week stage in both variants using glycerol and glucose, reaching 30% and 62% at the four-week growth stage, respectively (Figure 5D,E).

As for CL, its level in both variants decreased to trace amounts starting from the third week of cultivation, and completely disappeared by the fourth week of cultivation in the culture using 2% glycerol (Figure 5D,E). PS, PA+PI, PC, and SL also disappeared in both samples at the end of the experiment (Figure 5E).

#### 3.2.2. The Membrane Lipid Profile of *E. magnusii* Culture upon CR

The growth under limiting substrate conditions resulted in the maximum accumulation of the membrane lipids at the one-week stage of cultivation (Figure 4A). It is noteworthy to say that upon CR, at the late growth stages (3–4 weeks), the total level of the lipid components doubled compared to that using glycerol and 2% glucose. Unlike the cultures using 2% substrates, the CL level at the four- week stage of cultivation was only about 4% (Figure 5E). However, PA increased more than twice compared to that at the 24h stage of cultivation. Additionally, the PS amount increased to 6.46% and the St level doubled (Figure 5E). Unlike the cultures using 2% substrates, the PC level four-fold decreased by the end of the experiment compared to that at the 24 h stage (Figure 5A,E).

### 3.3. FAs of the Main PLs

#### 3.3.1. FAs Composition and the Degree of Unsaturation of Total PLs upon Cultivating *E. magnusii* Cells Using 2% Glucose and 2% Glycerol (2% Glucose vs. 2% Glycerol)

To determine the degree of unsaturation, the total PL fractions was chromatographically isolated, and their FAs composition was analyzed. Figure 6 shows the changes in the FAs composition and the degree of FAs unsaturation of the *E. magnusii* cell membranes. The palmitic acid C16:0 (10–23%), palmitoleic acid C16:1 (4.3–15.9%), oleic acid C18:1 (23.4–59.2%), and linoleic acid C18:2 (10.8–49.2%) were dominating in the FAs (Figure 6A–E). It is worth noting that the levels of the minor FAs (margarine acid C17:0, heptadecaenoic acid C17:1, and stearic acid C18:0) significantly (from 1 to 3–6%) increased in the PLs at the three- and four-week stages (Figure 6A–E). Some important features for the cultures on glycerol compared to that on 2% glucose were noted: the risen amount of monoenic C16:1 (palmitoleic) acid at the 24 h and four-week growth stages with simultaneous reduced level of dienic C18:2 (linoleic) acid (Figure 6A,E). In both cultures, at the four- week growth stage, there appeared some minor fractions of C12:0 (lauric acid) and C14:0 (myristic acid) in nearly the same concentrations (1.5–2.3%) (Figure 6E).

The degree of FAs unsaturation in the cultures grown using two substrates varied in different ways. The culture on 2% glucose showed two peaks at the beginning of the experiment (24 h of growth and in one week of cultivation), while upon assimilating glycerol, the degree of unsaturation reached its maximum at the one-week growth stage followed by an nearly two-fold decrease in it at the four-week stage (Figure 6F).

#### 3.3.2. FAs Composition and the Degree of Unsaturation of the PLs upon Cultivating the *E. magnusii* Cells under the CR Conditions

Some FAs, namely C16:0 (up to 20.6%), C18:1 (up to 43.3%), C18:2 (up to 56.4%), and C 16:1 (2.9–7.8%) dominated in the FAs fractions upon the CR (Figure 6A–E). At the four-week stage, a fraction of C18:3n3 (α-linolenic acid) increased its share in the phospholipids (up to 36%), while the fraction of C18:1 nearly completely disappeared (Figure 6E). Moreover, the fatty acid composition was not very diverse, namely there were no fractions of myristic acid C14:0, penta-decanoic acid C15:0, margarine C17:0, and hepta-decaenoic C17:1 acids. The C18:1 totally disappeared by the end of the experiment (Figure 6E).

The degree of unsaturation of FAs of the PLs upon of limited nutrition varied little throughout the whole experiment, reaching more than 1 and four-fold decreasing by the fourth week of cultivation (Figure 6F).

### 3.4. Changes in the Lipid Profile upon Carbohydrate Depletion

#### 3.4.1. Storage Lipids Profile

The total content of the storage lipids in the cells grown upon substrate depletion of 0.2% glucose at the one-week growth stage could be compared to that in the culture using glycerol (56.79 mg/g, *w*/*w* vs. 77.72 mg/g, *w*/*w*). At the three-week stage its three-fold increased was comparable with the data obtained for 2% and 0.5% glucose (Figure 7B). The two-week stage, where the amount of storage lipids halved compared to that on glycerol and 0.5% glucose and was three-fold lower than that on 2% glucose, was the only exception (Figure 7B). A tendency to four-fold increase in the storage lipids amount during the long-lasting cultivation of the *E. magnusii* cells seemed interesting and unexpected (Figure 7B).

The storage lipids composition included rather high amount of DAGs (about 30%) at the one-week stage and an insignificant share of TAGs, which reached no more than 15% with a high share of FFAs (Figure 7A). The trend remained throughout the whole experiment. Noteworthy to say, the lipid components of Y1 and Y2, found in the cells upon substrate depletion, showed quite a high share in the total lipid amount. They reached 20% or more and about 10%, respectively (Figure 7A).

#### 3.4.2. Membrane Lipids Profile

The cultivation of the yeast upon substrate depletion (0.2% glucose) led to a decrease in the total pool of membrane lipids at the one-week stage. It confirmed the general patterns obtained in the cells using 2% glucose and 2% glycerol (Figure 8B). We found a rather high level of these lipid components, compared to those on 2% substrates, and at the end of the experiment, the amount of membrane lipids even exceeded the values (17.11 mg/g, *w*/*w* vs. 14.66 and 12.71 mg/g, *w*/*w* in the cultures using 2% glycerol and 2% glucose, respectively) (Figure 8B).

The qualitative and quantitative composition of membrane lipids upon substrate depletion showed some features. (1) High level of PC at the four-week growth stage (25.49% vs. 6.46% upon CR) (Figure 8A). (2) High level of LPE + PI at the two- and three-week stages, exceeding 1.5–4.4 times those upon CR of 0.5% glucose and 1.6–13 times those on glycerol (Figure 8A); (3) Consistently high PA amount (24–33%) throughout the cultivation. (4) High SL amount at the four-week stage (8.5% vs. 0 for glycerol and 3.6% in the CR variant) (Figure 8A).

#### 3.4.3. FAs Composition of the Main Phospholipids

The fatty acid composition of the phospholipids of the *E. magnusii* cells included three major fractions: C16:0 (up to 20%), C18:1 (up to 15%), and C18:2 (up to 68.6%) (Figure 9). At the same time, the linoleic acid level at the 24 h growth stage was much higher than that in all the variants studied. The FAs phospholipid unsaturation index upon the substrate depletion reached 1.3–1.5 (Figure 9, cut-in) throughout the cultivation.

### 3.5. Visualization of Lysosomes Using E. magnusii LysoTracker Red DND99 Cells

The yeast culture is known to show a decrease in the anabolic processes activity as a mechanism of survival and energy saving in response to starvation. However, the lytic activity in lysosomes rises to degrade and facilitate the catabolic processes of the cell supply [27]. The lysosomes are the main cell organelles where there is hydrolysis of various macromolecules. They demonstrate a high concentration of protons and have more than 50 hydrolases with an optimal pH below 6.36 [28].

To detect these organelles in the cells, some specific probes based on weak-base amines are used, which selectively accumulate in the cell compartments with low pH and can be used to study the biosynthesis and pathogenesis of lysosomes [29]. LysoTracker probes consisting of a fluorophore bound to a weak-base are only partially protonated at neutral pH; they freely penetrate cell membranes and accumulate in spherical cell organelles. To assay the lysosomal cell activity of *E. magnusii* under various cultivation conditions, in vivo staining of the cells with a specific LysoTracker Red DND 99 probe was performed. The cells using 1% glycerol showed weak staining of the cytosolic wall without visible fluorescence enhancement in spherical organelles throughout the whole experiment (Figure 10A–D, (a)). At the four-week growth stage, there were single stained cells in the field (Figure 10D, (a)). However, nearly all the cells using 2% glucose showed high fluorescence at the four-week (Figure 10C,D, (b)). Upon calorie restriction, spherical areas in the cells considered as lysosomal structures were clearly seen at the one-week stage of cultivation (Figure 10B, (c)), whereas in the culture using 0.2% glucose, there were about 50% of these cells at the end of the experiment (Figure 10D, (b)).

### 3.6. ROS Level in the Glycerol and Glucose Assimilating E. magnusii Yeasts

The accumulation of sufficient amount of carbohydrates (both trehalose and glycogen) and the ROS elimination during the transition from the logarithmic to the stationary growth stage are the key factors of cell survival in the stationary growth phase [30]. The signaling pathways, along with fasting signals, include Yak1, Rim15, and Mck1 kinases, being negatively regulated by TOR and/or PKA. They control intracellular ROS level [31], which reflects the metabolic readjustment to increase the energy accumulation and to induce the antioxidant system, regulated by the key signal protein for survival [30].

The dynamics of the ROS level in the *E. magnusii* cells showed that the total quantity of oxygen radicals was significantly higher upon calorie restriction (Figure 11). Moreover, at the two-week growth stage in the CR and substrate depletion variants, the ROS level was 2.5 and 2 times higher, respectively, than the same one on glycerol (Figure 11). It should be noted that in the late growth stages the ROS amount in the cells upon CR-decreased, however, remained 1.5–1.75 times higher than that in glycerol-utilizing cells.

## 4. Discussion

During the growth, the yeast culture is affected by numerous factors, however, the most significant one is the restriction of the substrate. The cellular and molecular mechanisms leading to the cell survival or aging upon the extreme nutrient restriction remain insufficiently studied. CR has been the oldest known strategy promoting health since Avicenna [32], however, the exact molecular mechanism of the phenomenon is still being investigated. Several papers describe that the positive effect of CR on life expectancy is associated with the induction of Sirtuin-1 (Sirt1) [33] and autophagy [34]. Moreover, CR is known to alter the signaling of insulin-like growth factor 1 (IGF-1) by inhibiting Forkhead Box (FOXO) proteins [35], promoting mitochondrial biogenesis in humans [36], and correcting the expression of some genes with the function of mitochondrial biogenesis [37]. CR also rejuvenizes the culture reducing the oxidative stress through Sirt3-dependent promotion of superoxide dismutase [38].

In this study, we tried to assay the effect of calorie restriction on the lipid profile of the *E. magnusii* yeast during the long-lasting cultivation. The yeast organisms are known to be capable of effective aerobic growth using glycerol, and the transition from the “fermentative” to “respiratory” growth requires a significant transcriptional rearrangement [15], which involves the genes encoding the respiratory chain, oxidative, osmotic, and general anti-stress responses, protein synthesis enzymes, glycerol assimilation, etc. [14]. Moreover, during the long-lasting cultivation using substrates of “oxidative” and “fermentation” types and upon glucose limitation, the cellular metabolism adapts not only to the carbon source, but also to any changes in its availability being the stress for cells.

*The adaptation of the lipid composition of the E. magnusii cells during long-lasting cultivation*. The *E. magnusii* cells had a high level of storage lipids, where TAGs and FFAs dominated (Figure 3). The TAG amount was high in all the samples and reached 80% of the total lipid content in the cells using 2% glycerol, and up to 70% in the cells, using 2% glucose (Figure 3). It could be associated with abundant lipid drops (LD) in the *E. magnusii* cells, which are able to form structural complexes with mitochondria and nucleus (Figure 1E). The protective role of storage lipids, for TAGs and unsaturated FAs in particular, is widely known. TAGs can inhibit the chronological aging of the yeast due to their accumulation in LDs. It permits to deposit their most unsaturated FAs by esterifying them into TAGs [39,40]. As unsaturated FAs are very sensitive to age-related oxidative damage, their deposition in TAGs can make LD the main ROS target. It reduces the oxidative damage to the macromolecules in any other cell compartments [39,40]. Moreover, TAGs prevent lipo-toxicity by maintaining lipid homeostasis and providing a high chronological and replicative potential of yeast [39]. Upon assimilating glycerol and 2% glucose, the ROS level was moderately low (Figure 11), and we could suppose that high TAGs level in those cells provided the protection against oxidative damage in the stationary growth phase.

The FFAs level during the cultivation on glycerol, starting from the one-week growth stage (Figure 3B), showed a gradual decrease throughout the experiment (Figure 3E), whereas upon using 2% glucose, the its maximum reached 25-35% at the three-, four-week growth stage (Figure 3D,E). Synthesis of FFAs in the yeast cells goes on three main pathways: (1) de novo synthesis; (2) hydrolysis of complex lipids and delipidation of proteins; and (3) from the external sources [41]. The metabolism of 2% glucose, induces FFAs de novo synthesis in the cytosol and mitochondria, followed by elongation and desaturation in the endoplasmic reticulum (EPR). The initial stage of FFAs synthesis is catalyzed by acetyl-CoA-carboxylase (cytosolic enzyme is encoded by *ACC1*, the mitochondrial one is by-*HFA1*). In this reaction, acetyl-CoA is carboxylated to form malonyl-CoA, which serves as a two-carbon unit in the successive FFAs synthesis reactions [41]. It agrees well with a drop in the TAGs amount in the cells using 2% glucose with a simultaneous increase in FFAs during the cultivation (Figure 3). On the contrary, upon glycerol assimilating FFAs are probably involved in the synthesis of TAGs and their accumulation in the composition of LD. The central precursors of TAGs are known to be PA synthesized from either glycerol-3-phosphate or dihydroxyacetone phosphate. Upon cleavage of PA with phosphatidate phosphatase Pah1p into diacylglycerols, they become direct precursors for the TAGs synthesis [41]. To form TAGs, the intermediate DAGs is acetylated at the sn-3 point. This stage can be performed in acyl-CoA-independent or acyl-CoA-dependent reactions [41]. Acyl-CoA-independent reaction catalyzed by phospholipid:diacylglycerol acyltransferase (Lro1) uses phospholipids as an acyl donor. Two Est synthases of Are1p and Are2p are capable of catalyzing the acyl-CoA-dependent formation of TAGs, though with less efficiency. Lro1p and two Est synthases are located in ER, whereas another acyltransferase (Dga1p) requiring oleoyl-CoA or palmitoyl-CoA as co-substrates is reported to be of double location, namely in LD and ER [41]. The decrease in the total amount of DAGs (Figure 3) and phospholipid components (Figure 4) at the late growth stages, as well as significant decrease in the PA pool up to its complete exhaustion in the culture using 2% glucose at the four-week growth stage (Figure 4) could indicate that all these components are likely to be involved in the TAGs synthesis, which provides high metabolic activity of the *E. magnusii* cells during the long-lasting cultivation.

The cell membranes, being a cell protective barrier, change their lipid composition in response to any external influence. PC, PE, PI, and PS are traditionally considered to be the main PLs in the *S. cerevisiae* extracts [41]. However, the carbon source used for yeast cultivation could significantly affect the PLs composition [42]. PE, St (more than 30% of the total lipid amount), PC, and PA (more than 40%) dominate in the membrane lipids of the *E. magnusii* cells (Figure 4). This composition differs greatly from the declared in the references composition for *S. cerevisiae* in the high St and PA level [43,44]. It should be noted that during our experiments, the St amount doubled at the three-week growth stage and reached 30% and 62% at the four-week one for glycerol and glucose, respectively (Figure 5D,E). Sts are nonpolar lipids protected by SL head groups inside the membranes. Ergosterol, the end product of a complex pathway of St biosynthesis, is embedded in the plasma membrane and participates in its integrity is the main St of *S. cerevisiae* [41]. Ergosterol, as the most important component of fungal cell membranes, determines the fluidity, permeability, and activity of membrane-bound proteins. The deficiency in the St biosynthesis in the yeast causes pleiotropic defects, limiting the cell proliferation and adaptation to stress [45]. Thus, the ergosterol level in the yeast is strictly regulated by the bioavailability of some metabolites (for example, St, oxygen, and iron) and some environmental conditions.

PA, another major component of the *E. magnusii* cell membrane lipids, serves as a central metabolite in the PL synthesis de novo. Its precursor, LPA, is formed from glycerol-3-phosphate (G3P), in a reaction catalyzed by G3P-acyltransferases (SCT1 (GAT2) and GPT2 (GAT1)) [41]. PA can participate in two main processes of lipid biosynthesis. (1) It can be broken down into diglycerol by phosphatidate phosphatase Pah1p, Dpp1p, Lpp1p, and App1p. Then, it is included in the lipid accumulation pathway [46]; (2) Another process is when it is involved in the PL synthesis through the conversion to cytidine diphosphate diacylglycerol (CDP-DG) catalyzed by CDP-DG synthase Cds1p localized in ER [47] or via mitochondrial CDP-DG synthase Tam41p [48] with cytidine triphosphate (CTP) as CDP donor. For *S. cerevisiae* yeast, the PA amount was determined at the level of about 1% of the cell total phospholipids [43,44], while we assay its share of nearly 30% at the two-week growth stage using 2% substrates (Figure 5C,D).

Our data agree well with the results by Yanutsevich et al. [49], which showed that in the mycelial fungus of *Aspergilus niger* the stressors of various nature caused significant increase in the PA amount in its membranes. The authors suggested that the PA contribute into the adaptation of the fungus to stress should be to increase the cell membranes stability and to intensify vesicular transport, endocytosis, and exocytosis. However, an increase in the share of St and SL in the membrane lipids composition was noted in the alkalophilic micromycete of *Sodiomyces tronii* under the acidic conditions [50]. The deviation of the pH values from optimal in the alkalophiles of *S. magadii* and *S. alkalinus*, as well as the alkalotolerants of *Acrostalagmus luteoalbus* and *Chordomyces antarcticus* led to an increase in the St and PA share in the membrane lipids [51]. Some studies have studied the effect of St composition on the survival of the *S. cerevisiae* culture at various stress. Thus, the survival rate of the erg6Δ mutant, unable to synthesize ergosterol, significantly decreased at osmotic shock [52]. Perhaps, the high level of St in our experiments also determines the stability of the membrane structures in the *E. magnusii* cells. Significant amount of PA at the deep stationary stage (two weeks of cultivation) determines the storage source of TAGs and membrane phospholipids for life support.

High CL amount within 5 to 9% at the two-, three-week growth stage of cultivation proved an important feature of the *E. magnusii* membrane lipids during long-lasting cultivation (Figure 5). Of note, it was 1.5–2 fold higher than that for *S. cerevisiae* [43,44]. CL performs numerous cellular functions.It is associated with all major proteins of the mitochondria respiratory chain, and thereby increases the efficacy of electron flow and ADP/ATP exchange [53], and modulates the catalytic activity and stability of interacting proteins [54]. CL is crucial for the biogenesis of mitochondrial proteins [55], as it promotes mitochondrial fission and fusion [56], and participates in the support of the crest morphology and structure [57]. We can conclude that at some growth stages, the high CL level helps yeast culture maintain the energy balance of mitochondria and provides cell survival under those conditions.

A high amount of unsaturated fatty acids, namely oleic C18:1 (up to 59.2%) and linoleic C18:2 (up to 49.2%), is another factor providing the high adaptability of the *E. magnusii* cells upon long-lasting cultivation (Figure 6A–E). There are convincing data of the effect of the fatty acid composition in the membrane lipids on the yeast cells resistance to some kinds of stress. For example, the increased oleic acid amount causes the *S. cerevisiae* resistance to ethanol. The increase the monounsaturated acid level altogether with a decrease in the linoleic and linolenic acids amount compensates for an increase in membrane fluidity at ethanol stress [58]. An increase in the cytoplasmic membrane fluidity due to an increase in the linoleic acid share also increased the *S. cerevisiae* yeast resistance to osmotic stress. The *S. cerevisiae* transformants with embedded delta(12)-fatty-acid desaturase FAD2 and acyl-lipid omega-3 desaturase FAD3 of sunflower in the genome demonstrated higher resistance to these stressors upon increased membrane fluidity [59]. However, an increase in the minor saturated FAs (margarine C17:0 and stearin C18:0) share provided the maintenance of some membrane rigidity (Figure 6F).

*Restriction on the substrate*. The analysis of the CR (0.5% glucose) effect on the storage lipids level showed some decrease in the TAGs amount due to an increase in FFAs (Figure 3A,E). It was consistent with the degradation of storage and complex lipids to compensate for the energy needs of the cells [41]. The TAGs and Est degradation from LD by acting TAG and Est lipases needs to replenish the FAs pool [60]. The TAG and Est mobilization could be necessary to quickly meet the cell’s need for sterols and fatty acids, considering that the Est level in the cultures grown upon CR did not decrease significantly. Unlike TAGs, it remained at the level of 12-15% throughout the whole experiment (Figure 3), the FAs replenishment happened due to the mobilization of TAGs. The TAGs hydrolysis by yeast lipases provides the need for fatty acids and DAGs for the biosynthesis of complex membrane lipids. The family of yeast TAG lipases includes Tgl3p, Tgl4p, and Tgl5p localized in LD [60]. Tgl3p is likely to be the main TAG lipase in the yeast, although all three lipases are involved in the TAGs catabolism in vivo.

The increase in the FAs amount was related to an increase in the membrane lipid level, which doubled compared to that on glycerol and 2% glucose at the late stationary growth stages (3–4 weeks) (Figure 4). Probably, the TAGs mobilization could serve as a means of membrane lipids pool maintaining upon CR. A decrease in TAGs amount could be associated with a transitive drop in the DAGs level (up to 16%) (Figure 3) at one-week stage of cultivation, which also confirms the switching off the degradation processes of storage lipids to the phospholipids synthesis. Upon analyzing the spectrum of the components at calorie restriction, we found some decrease in the PC and PE level and simultaneous significant increase in the PA (up to 43%) and St (up to 19%) (Figure 6) shares, which play a protective role under these conditions [49,50]. At the four-week growth stage, the CL level was about 4%, while in the cultures using 2% substrates, it disappeared (Figure 5E), which may also be a factor determining the adaptation to nutrient restriction [61].

C16:0, C16:1, C18:1, and C18:2 dominated in the fractions of FAs upon CR (Figure 6A–E). Of note, at the four-week stage, a fraction of polyunsaturated C18:3n3 α-linolenic acid appeared at high level (up to 36%), whereas oleic acid almost completely disappeared (Figure 6E), and the overall degree of FAs unsaturation of FAs decreased (Figure 6F). The composition of its membrane fatty acids, containing only saturated and monounsaturated FAs and no polyunsaturated FAs, is the peculiarity of the *S. cerevisiae* yeast [42]. It gives the yeast membranes high resistance to lipid peroxidation, since the saturated FAs and monounsaturated FAs are not susceptible to free radical attack, because there are no bis-allylic hydrogens. Currently, the synthesis of polyunsaturated fatty acids is known to be performed by five different strains of the *Schizochytrium* species of the *Thraustochytrids* group [62,63] and *Fusarium verticillioides NKF1*, producing C18:3ω3 [64]. Taking all the data together, we can assume that the *E. magnusii* yeast is capable of producing the polyunsaturated FAs, and this is an important adaptive feature of this yeast when nutrition is restricted.

*Depletion of the substrate*. The effect of depletion of the substrate (0.2% glucose) shown as: (1) high total amount of storage and membrane lipids; (2) a trend to a significant increase in the storage lipids amount during growth; (3) a high PA share throughout the whole growth; and (4) a steadily increased degree of unsaturation throughout the experiment. It is noteworthy that in the culture using 0.5% glucose TAGs and FFA were the main components of the storage lipids and the FA composition of *E. magnusii* included C16:0 (up to 20%), C18:1 (up to 15%), and C18:2 (up to 68.6%) (Figure 9). However, the amount of the linoleic acid at the stationary stage was higher than in all versions tested. The linoleic acid (C18:2) is a polyunsaturated FA n-6 involved in forming lipids during the metabolism, and has a positive effect in the treatment of many human diseases, such as atherosclerosis, inflammation, and carcinogenesis [65]. The polyunsaturated FAs in the yeast cells are formed from the stearic acid (C18:0). It is converted by the action of Δ9-desaturase (Ole1) into the oleic acid, which, in turn, can subsequently be converted into the linoleic acid using Δ12-desaturase (FAD2) and into the α-linolenic acid using Δ15-desaturase (FAD3) [66]. We can suppose that the redistribution of the synthesis towards the polyunsaturated fatty acids is a factor providing high adaptability upon the substrate depletion.

*Substrate restriction, ROS generation and autophagy*. The process of autophagy is known to be obligatory when the nutrients are depleted to supply the cell with building blocks from the degradated macromolecules. The importance of autophagy is emphasized by the fact that the autophagy is of a strictly regulated nature, where several regulatory components, including TORC1, PKA, Snf1, Sch9, and Rim15, are involved in the active autophagic activity [67]. Thus, both TORC1- and PKA-mediated phosphorylation of Atg13 are reported to inhibit autophagy whereas upon substrate excess some growth-stimulating processes are activated. We showed that upon calorie restriction the autophagic processes were observed at the one-week growth stage. At the end of the experiment, in the culture using 0.2% glucose the cells containing autophagosomes made up about 50% (Figure 10D, (b)). At the late stationary stages upon calorie restriction, the ROS level remained 1.5–1.75 times higher than that in glycerol-utilizing cells (Figure 11). The flexibility of metabolic readjustment on the one hand, and the balance between energy, oxygen radical levels and stress resistance in the yeast on the other hand, are the fundamental base of survival and prolongation of life during the cultivation [31]. The hypothesis has two main principles: (1) the yeast strains with reduced TOR signaling demonstrate the increased mitochondrial ROS generation during the active growth, which decreases their level in the stationary phase, prolonging life span under the conditions [68]. The hormesis induced by mtROS and the increase in life span include promotion of the stress response; (2) catalase inhibition increases chronological lifespan by increasing the level in hydrogen peroxide, which induces superoxide dismutase to block superoxide accumulation [69]. If the postulates are taken into account, we can conclude that decrease in the nutrients availability, on the one hand, triggers TORC1 signals and autophagy, and on the other hand, maintains a relatively high level of ROS ensuring high life span in these conditions.

*Possible applications of CR in biotechnology*. We have shown that upon calorie restriction, the *E. magnusii* yeast get the ability to increase the polyunsaturated fatty acids synthesis. The linoleic acid level reaches 36%, and the α-linolenic acid one does up to 68.6%. The high levels of the unsaturated fatty acids were shown for both oleaginous yeasts, which can accumulate abundant amount of lipids (*Yarrowia lipolytica, Rhodotorula glutinis, Trichosporon cutaneum,* and *Candida* sp.), and the traditional nonoleaginous yeast (*Kluyveromyces polysporus, Torulaspora delbrueckii,* and *Saccharomyces cerevisiae*) if the nitrogen source and its availability change [70]. At the same time, the linoleic acid level in *Kluyveromyces* and *Trichosporon* strains reached nearly 50% of the total amount of fatty acids upon the long-lasting cultivation and nitrogen starvation. *Saccharomyces occidentalis* showed the increased production of the linoleic acid from 5.6% to about 23% upon overexpression of FAD2 [71]. Perhaps, the application of CR as a regulation factor of the lipid metabolism could be helpful for technological schemes altogether with some other methodological approaches.

## Figures and Tables

**Figure 1 jof-08-01233-f001:**
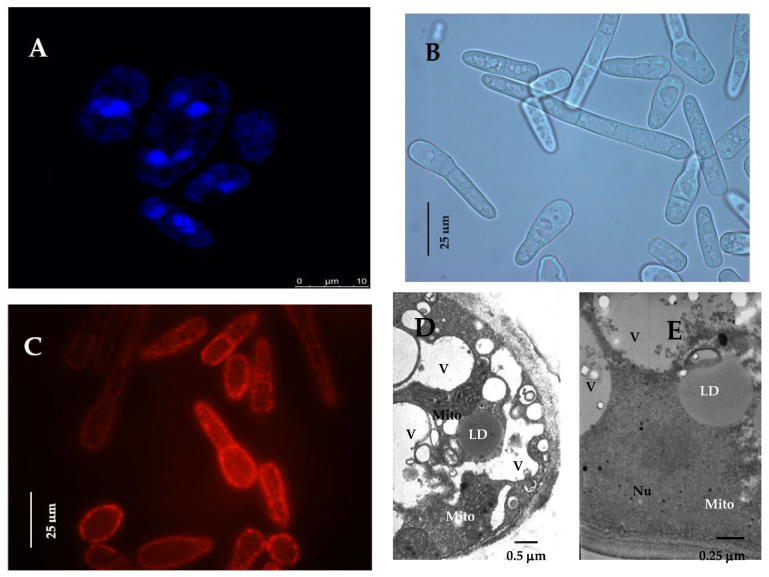
Micro images of the *E. magnusii* cells in the logarithmic growth stages. (**A**) Fluorescent micro-images of the cells, labeled with 0.3 μM DAPI for DNA; (**B**) mycelial form of cells; (**C**) the potential-dependent stain of the mitochondria in *E. magnusii* cells by 5 mM Rh123. The regions of high mitochondrial polarization are bright red due to the concentrated dye. The cells were examined after 30 min. The incubation medium contained 0.01 M phosphate-buffered saline (PBS) and 1% glycerol, with a pH of 7.4. To examine the Rh123-stained preparations, filters 02, 15 (Zeiss) were used (magnification 100×). Photos were taken using an AxioCam MRc camera: (**D**,**E**)-ultrastructure of *E. magnusii* cells; Nu—nucleus; V—vacuole; Mito–mitochondria; LD—lipid drops.

**Figure 2 jof-08-01233-f002:**
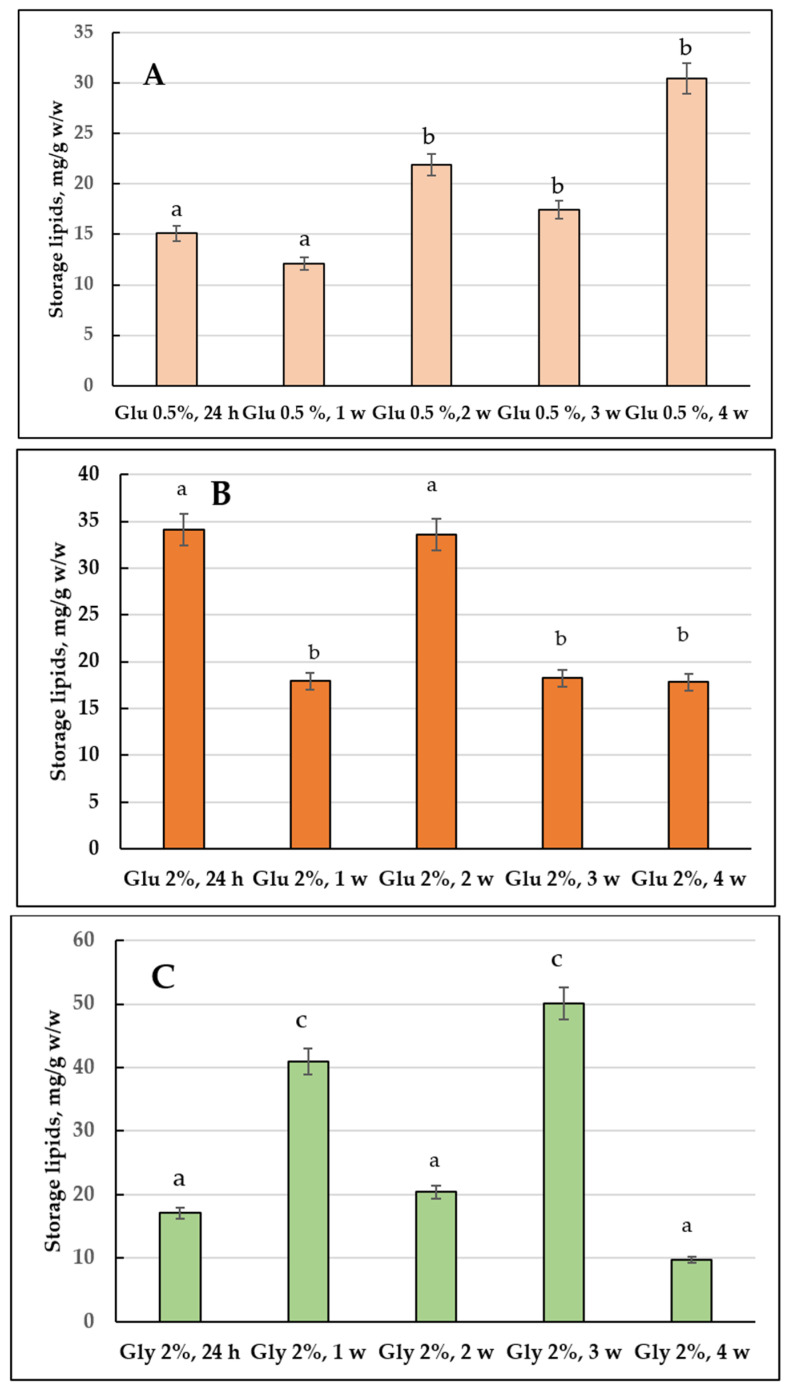
The storage lipid composition in the *E.magnusii* yeast grown on the different substrates. The share of each storage lipid fraction, %; (**A**) glucose 0.5%; (**B**) glucose 2%; (**C**) glycerol 2%. Error bars represent the standard deviation of triplicates. Mean values are displayed (n = 3, ±SD). a—*p* < 0.04; b—*p* < 0.01; c—*p* < 0.02.

**Figure 3 jof-08-01233-f003:**
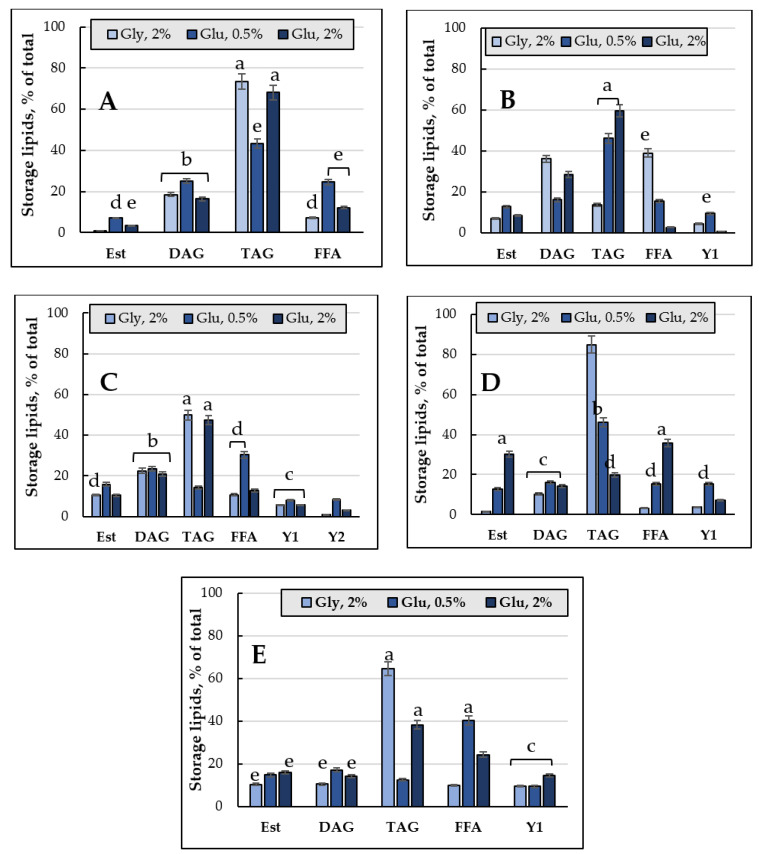
The total storage lipids content, mg/g *w*/*w*. TAGs—Triacylglycerols; DAGs—Diacylglycerols; FFAs—Free fatty acids; Est—sterol esters. (**A**) 24 h of cultivation; (**B**) 1 week; (**C**) 2 weeks; (**D**) 3 weeks; (**E**) 4 weeks. Error bars represent the standard deviation of triplicates. Mean values are displayed (n = 3, ±SD): a—*p* < 0.04; b—*p* < 0.01; c—*p* < 0.001; d—*p* < 0.02; e—*p* < 0.002. The means without letters did not differ significantly.

**Figure 4 jof-08-01233-f004:**
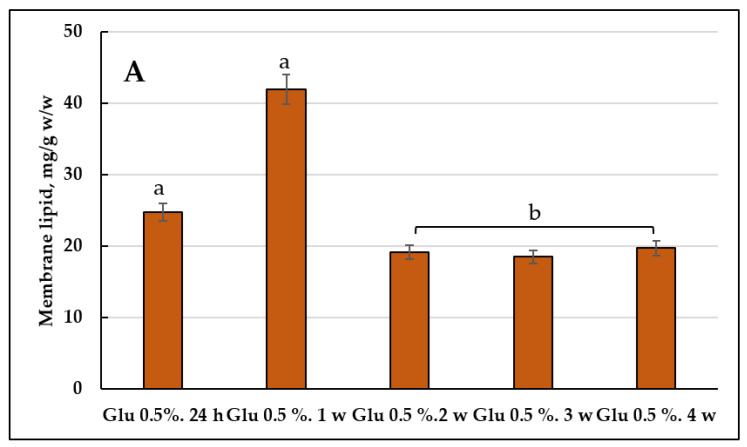
The membrane lipid composition in the *E.magnusii* yeast grown on the different substrates. The share of each storage lipid fraction, %; (**A**) glucose 0.5%; (**B**) glucose 2%; (**C**) glycerol 2%. Error bars represent the standard deviation of triplicates. Mean values are displayed (n = 3, ±SD). a—*p* < 0.04; b—*p* < 0.01; c—did not differ significantly.

**Figure 5 jof-08-01233-f005:**
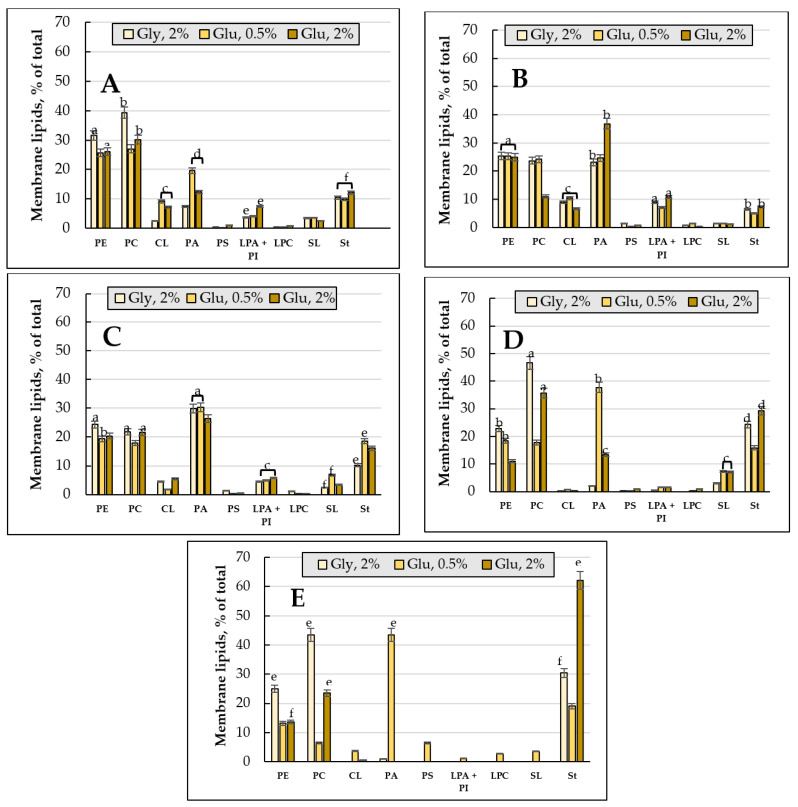
Membrane lipid composition of *E.magnusii* under different conditions. The share of each membrane lipid fraction, %; (**A**) 24 h of cultivation; (**B**) 1 week; (**C**) 2 weeks; (**D**) 3 weeks; (**E**) 4 weeks. PE—Phosphatidylethanolamines; PC—Phosphatidylcholines; CL—Cardiolipins; PA—Phosphatidic acids; LPA + PI—lysophosphatidylethanolamine + Phosphatidylinositols; LPC—Lysophosphatidylcholines; SL—Sphingolipids; St—Sterols. The conditions of the culture growth are indicated in the panels. Error bars represent the standard deviation of triplicates. Mean values are displayed (n = 3, ±SD). a—*p* < 0.04; b—*p* < 0.01; c—*p* < 0.001; d, e—*p* < 0.02; f—p < 0.002. The means without letters did not differ significantly.

**Figure 6 jof-08-01233-f006:**
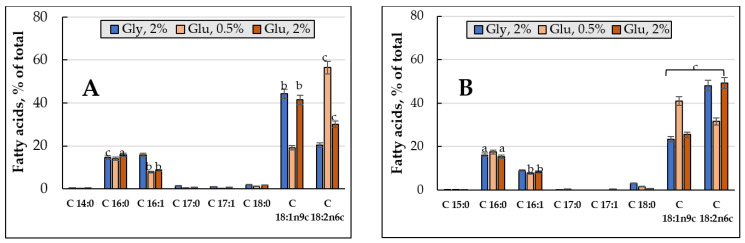
Fatty acid composition of the main membrane phospholipids of *E.magnusii* (% of total amount). (**A**) 24 h of cultivation; (**B**) 1 week; (**C**) 2 weeks; (**D**) 3 weeks; (**E**) 4 weeks; (**F**) the unsaturation degree. Mean values are displayed (n = 3, ±SD); a—*p* < 0.04; b—*p* < 0.01; c—*p* < 0.001; d—*p* < 0.02; e—*p* < 0.02. The means without letters did not differ significantly.

**Figure 7 jof-08-01233-f007:**
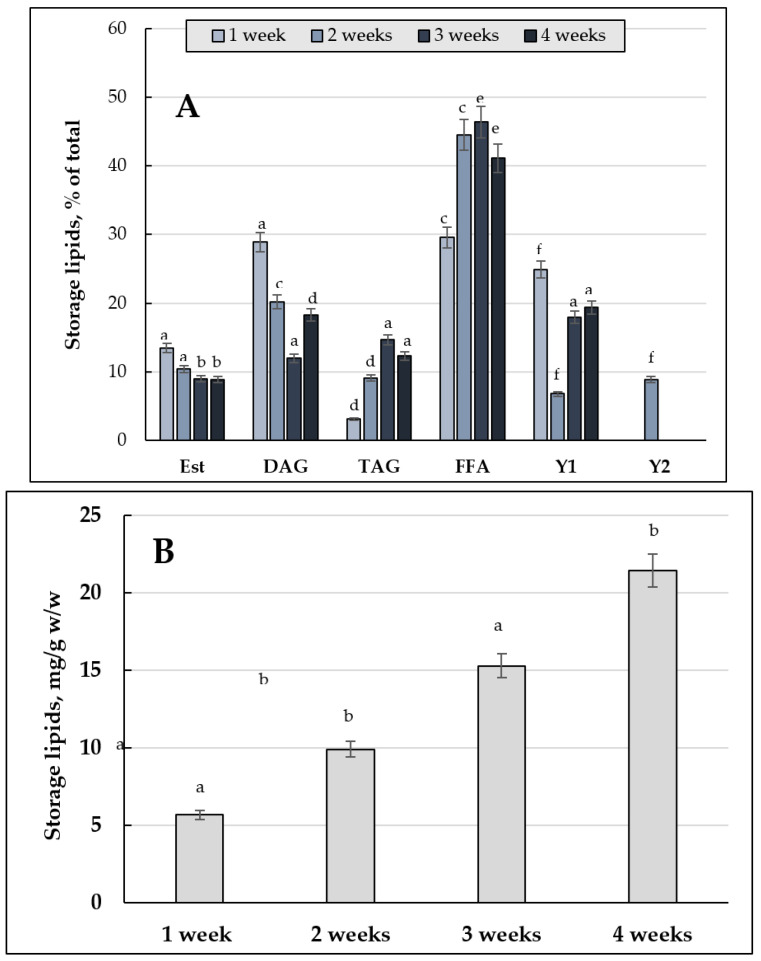
The storage lipid composition in *E. magnusii* yeast in the different growth phases during long-lasting cultivation under 0.2% glucose utilization. (**A**) the share of each storage lipid fraction, %; TAGs—triacylglycerols; DAGs—diacylglycerols; FFAs—free fatty acids; Est—sterol esters; Y1, Y2—unknown fractions (**B**) the share of each storage lipid fraction (μg/g). Values are mean ± S.E.M from three independent experiments and three analytical replicates. a—*p* < 0.04; b—*p* < 0.01; c—*p* < 0.001; d, e—*p* < 0.02; f—*p* < 0.002.

**Figure 8 jof-08-01233-f008:**
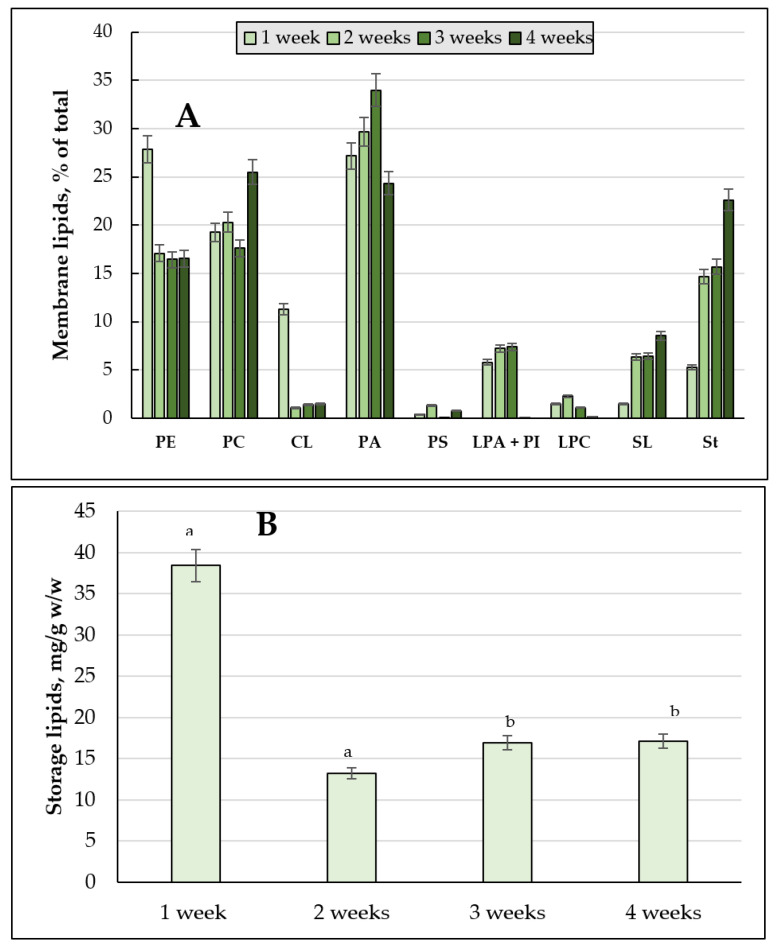
Membrane lipid composition of the mitochondria from *E. magnusii* cells raised in the different growth phases during long-lasting cultivation under 0.2% glucose utilization. (**A**) the share of each membrane lipid fraction; PE—phosphatidylethanolamines; PC—phosphatidylcholines; CL—cardiolipins; PS—phosphatidylserine; LPE +PI—lysophosphatidylethanolamine + phosphatidylinositols; LPC—lysophosphatidylcholines; SL—sphingolipids; St—sterols; X1, X2—unknown; (**B**) the total membrane lipid content. The means in panel A did not differ significantly. a—*p* ≥ 0.04; b—*p* ≥ 0.05.

**Figure 9 jof-08-01233-f009:**
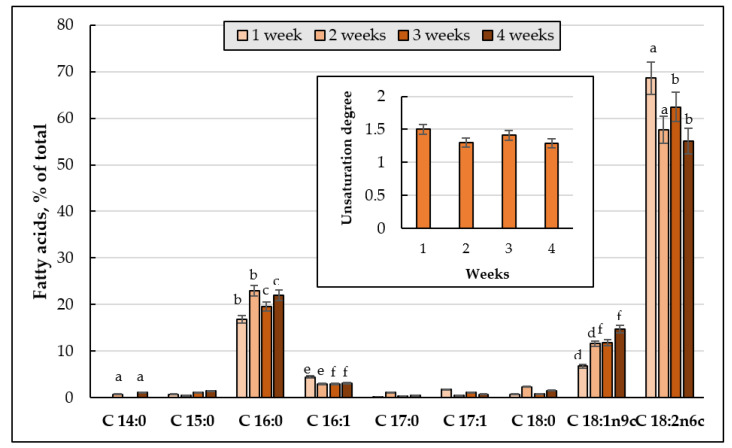
Fatty acid composition of the main membrane phospholipids of *E.magnusii* (% of total amount) using 0.2% glucose. In the cut-in, there is the degree of unsaturation of FAs of the phospholipids. Mean values are displayed (n = 3, ±SD). a—*p* < 0.04; b—*p* < 0.01; c—*p* < 0.001; d, e—*p* < 0.02; f—*p* < 0.002.

**Figure 10 jof-08-01233-f010:**
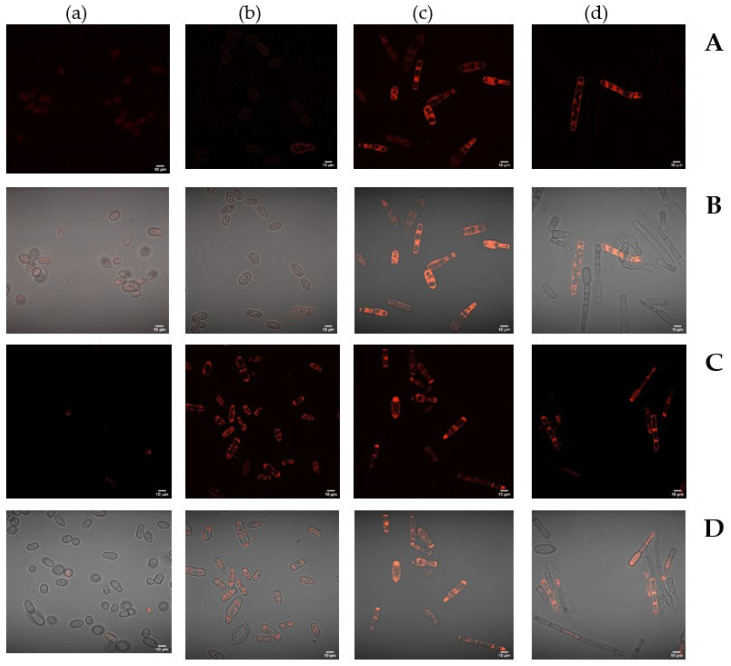
The visualization of the lysosomes after staining with LysoTracker Red DND-99. (**A**,**B**) is one-week stage; (**C**,**D**) four-week stage. (**A**,**C**) Fluorescent microscopy; (**B**,**D**) optical and fluorescent images. (**a**)—glycerol 2%; (**b**)—glucose 2%; (**c**)—glucose 0.5%; (**d**)—glucose 0.2%.

**Figure 11 jof-08-01233-f011:**
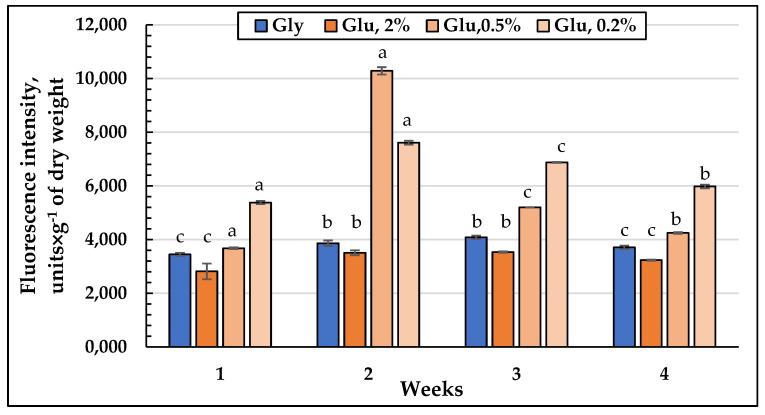
ROS level in the glycerol and glucose assimilating *E. magnusii* yeast, raised at different growth stages. The dynamics of intracellular ROS production was monitored using a spectroscopic fluorescence probe of dihydro-2′,7′-dichlorofluorescein diacetate ester H_2_DCF-DA. Values are mean ±S.E.M from 5–6 independent experiments: a–c—0.05 > *p* > 0.005. The means without letters did not differ significantly.

## Data Availability

Not applicable.

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
