# Peer review of "The Lipid Profile of the Endomyces magnusii Yeast upon the Assimilation of the Substrates of Different Types and upon Calorie Restriction"

_jof, 2022, doi:10.3390/jof8111233_

Round 1

Reviewer 1 Report

Please check the attached file

Author Response

Dear Reviewer!

Our answers are in attached file.

Sincerely yours, the Authors

Reviewer 2 Report

Review of the manuscript entitled ‘The lipid profile of the Endomyces magnusii yeast upon the assimilation of the substrates of different types and upon calorie  restriction’ submitted to the  Journal of Fungi.

1. The abstract is way too long. It should be shortened twice.

2. 2.1 - What is the origin of the strain? Is it a collection strain? If so please provide numbers. If isolate - how was it identified? Please provide your GenBank excise numer.

3. ‘To determine cell lipids, yeast cells in the stationary growth phase’ - specify the time of cultivation

4. ‘Analysis of soluble carbohydrates and lipids was 218 performed using one-way ANONA’ - were the normal distribution and homogeneity of variance checked?

5. 3.1. - giving results in the form of µg/g is illegible. Either change to mg/g, or convert as most researchers to g/100 g

6. Why is page 8 practically empty?

Apart from these minor remarks, I found the article very interesting and well written. After correcting the manuscript, I recommend its publication in the Journal of Fungi.

Author Response

Dear Reviewer!

Our comments and the answers are in attached file.

Sincerely yours, the Authors
